

# A comparison of a two-dimensional depth averaged flow model and a three-dimensional RANS model for predicting tsunami inundation and fluid forces

Xinsheng Qin[1], Michael Motley[1], Randall LeVeque[2], Frank Gonzalez[3], and Kaspar Mueller[4]

[1]Department of Civil and Environmental Engineering, University of Washington, More Hall Box 352700, Seattle, WA 98195
[2]Department of Applied Mathematics, University of Washington, Seattle, WA 98195
[3]Department of Earth and Space Sciences, University of Washington, Seattle, WA 98195
[4]School of Computer Science and Communication, KTH, Royal Institute of Technology, SE-100 44 Stockholm, Sweden

**Correspondence:** Xinsheng Qin (xsqin@uw.edu)

**Abstract.** The numerical modeling of tsunami inundation that incorporates the built environment of coastal communities is challenging for both depth-integrated 2D and 3D models, not only in modeling the flow, but also in predicting forces on coastal structures. For depth-integrated 2D models, inundation and flooding in this region can be very complex with variation in the vertical direction caused by wave breaking on shore and interactions with the built environment and the model may not be able to produce enough detail. For 3D models, a very fine mesh is required to properly capture the physics, dramatically increasing the computational cost and rendering impractical the modeling of some problems. In this paper, comparisons are made between GeoClaw, a depth-integrated 2D model based on the nonlinear shallow water equations (NSWE), and OpenFOAM, a 3D model based on Reynolds Averaged Navier-Stokes (RANS) equation for tsunami inundation modeling. The two models were first validated against existing experimental data of a bore impinging onto a single square column. Then they were used to simulate tsunami inundation of a physical model of Seaside, Oregon. The resulting flow parameters from the models are compared and discussed, and these results are used to extrapolate tsunami-induced force predictions. It was found that the 2D model did not accurately capture the important details of the flow near initial impact due to the transiency and large vertical variation of the flow. Tuning the drag coefficient of the 2D model worked well to predict tsunami forces on structures in simple cases but this approach was not always reliable in complicated cases. The 3D model was able to capture transient characteristic of the flow, but at a much higher computational cost; it was found this cost can be alleviated by subdividing the region into reasonably sized subdomains without loss of accuracy in critical regions.





# 1 Introduction

For many years, researchers have been working on different numerical models that can predict tsunami behavior. Tsunami prediction generally requires modeling at a wide range of spatial scales, including (from large to small scale): offshore wave propagation, beach runup, inland inundation, and impact on individual structures.

Due to the large differences in scale for the different processes, most tsunami models solve two-dimensional depth-integrated equations, e.g., the nonlinear shallow water equations (NSWE) or some form of Boussinesq wave equations to predict tsunami behavior, using computational grids that vary several orders of magnitude in spatial resolution, from several kilometers far from the shoreline to 10 meters inland. The NSWE are often used in the nearshore and inundation zone, since they can handle nonlinearities that arise in very shallow water and can be adpated to deal robustly with wetting and drying. However, it is

not clear that these equations are adequate to properly model fully three-dimensional turbulent flow, particularly at the scale necessary to determine tsunami impact and corresponding tsunami-induced forces on individual structures.

    It would be preferable to solve the three-dimensional Navier-Stokes equations with a proper turbulence closure. However, this is still extremely expensive computationally relative to two-dimensional models, and only practical for detailed simulations over small spatial regions.

The scale of modeling inland tsunami inundation with an explicitly represented constructed environment lies between that of modeling the large-scale tsunami wave propagation offshore and the small-scale tsunami impact on individual structures. This process is actually even more challenging to model since for two-dimensional depth-integrated models, inclusion of the constructed environment increases the complexity of the topography and the flow begins to have more variation in the vertical direction, while for the three-dimensional model that solves the Navier-Stokes equations, a fine mesh needs to be generated

around each individual structure, which dramatically increases the number of cells in the computational domain.

    In this paper, we compare results from a two-dimensional NSWE model and a 3D Navier-Stokes model for the test case of flow through a scale model of a portion of Seaside, Oregon. The experiment was performed in the directional wave basin at the O.H. Hinsdale Wave Research Laboratory at Oregon State University and produced a large set of observed data of flow depth and velocities, as well as corresponding momentum flux, at many locations in the model Park et al. (2013). We use two

open source models, the 2D GeoClaw software from Clawpack Clawpack Development Team (2015), which is widely used for modeling tsunamis (both global propagation and local inundation), and the 3D OpenFOAM software (The OpenFOAM Foundation, 2014). The two models are first compared and validated against an experiment in which a simple bore impinges on a single column, and then compared for the Seaside model. The goal is to explore the differences between 2D and 3D modeling for this complex case, and to provide some guidance for modeling tsunamis or other flooding events in similar constructed

environments.

    Before introducing the two numerical models used in current study, a brief review of previous research involving different types of models is given below.

    The two-dimensional depth-integrated equations are the most widely used tsunami models for their simplicity and computational efficiency. Popinet (2012) simulated the 2011 Tohoku tsunami by solving the 2D NSWE with dynamically-adapted





spatial resolution that varied from 250 m in flooded areas nearshore up to 250 km offshore. The model accurately predicted long-distance wave and coarse-scale flooding; the initial surface elevation was determined from a source model based on seismic inversion (as opposed to inversion of DART buoys and tidal gauge time series). This also showed that an accurate and consistent model of tsunami wave propagation can sometimes be constructed using only seismic wave inversion. Wei et al. (2013) used the Method of Splitting Tsunamis (MOST) model to simulate the same tsunami event. The MOST model solves the shallow water equations in spherical coordinates with numerical dispersion. Their results demonstrated that it may be possible to forecast near-field tsunami inundation in real time. Hu et al. (2000) presented an NSWE model that can simulate storm waves propagating in the coastal surf zone and overtopping a sea wall. They found that waves overtopping a vertical wall may be approximately modeled by representing the wall as a steep slope, and that the overtopping rate is sensitive to the bottom friction and the minimum friction depth. The two-dimensional NSWE model of wave run-up and overtopping by Hubbard and Dodd (2002) features an adaptive mesh refinement algorithm. Their model can accurately reproduce 1D and 2D wave transformation, run-up and overtopping in physical experiments. Their modeling of seawall overtopping by off-normal incident waves showed that there can be more flooding in such a situation than at normal incidence. Lynett (2007) simulated long wave runup obstructed by an obstacle and concluded that the obstacle can help reduce runup and maximum overland velocity if the wave is highly nonlinear (with a ratio of wave height to shelf water depth $\geq 0.5$). The sensitivity study also showed that in cases of breaking waves, the Boussinesq model was more accurate than the nonlinear shallow water equations in terms of wave runup (maximum differences up to $10\%$). For nonbreaking long waves, differences between the two were negligible. Shi et al. (2012) developed a high-order adaptive time-stepping TVD solver for a fully nonlinear Boussinesq model and validated it against a series of laboratory experiments for wave shoaling and breaking and a suite of benchmark tests for wave runup. The results showed that the model was able to accurately model wave shoaling, breaking, and wave-induced nearshore circulation. With a Boussinesq model, Lynett et al. (2010) simulated overtopping of levees of the Mississippi River-Gulf Outlet (MRGO) during Hurricane Katrina at four characteristic transects along the 20 km-long stretch of the levees. The predicted overtopping rates agreed well with the observed data.

As computing power increases, it becomes possible to model the tsunami runup process, instead of simply wave impact on an individual structure, by solving three-dimensional Navier-Stokes equations with a proper turbulence closure. Choi et al. (2007) solved three-dimensional Reynolds Averaged Navier-Stokes (RANS) equations to simulate wave runup on an conical island and compared different turbulence closure models including $k - \epsilon$ , RNG (Re-Normalisation Group methods, (Yakhot et al., 1992)) $k - \epsilon$ and LES (Large Eddy Simulation). Their results showed that LES and RNG $k - \epsilon$ are similar and more accurate than $k - \epsilon$ is worse than those two. Williams and Fuhrman (2016) solved incompressible RANS equations with a transitional variant of the standard two-equation $k - \omega$ turbulence closure to study boundary layer flow induced by tsunami-scale waves. Their results indicated that the boundary layer generated by a tsunami is both current-like due to the long duration and wave-like due to its unsteadiness. The study also indicated that an existing expression for maximum bed shear stress under wind wave scale can be reasonably extrapolated to full tsunami scale. Mayer and Madsen (2000) investigated wave breaking in the surf zone by solving the RANS equations with a $k - \omega$ turbulence model. They found that the volume-of-fluid method could





be used successfully to simulate wave breaking and that although some instabilities occurred in applying the RANS equations, they can be eliminated by an ad-hoc modification of the turbulence model.

The prediction of tsunami impact on individual structures is also important because it provides guidance on designing coastal structures in tsunami inundation zones. The two-dimensional depth-integrated model may not work properly for these scenarios since the problems are more three-dimensional with large variation in the vertical direction and with transient and turbulent flow impacting the structure. In these cases, a three-dimensional model that solves the Navier-Stokes equation may give much better results. Researchers at University of Washington modeled a series of "dam break" experiments by solving the 3D Reynolds Averaged Navier-Stokes (RANS) equations for bore-type impact of a wave on a series of 1/20-scale model girder bridges to assess the 3D effects on bridge skew (Motley et al., 2015; Wong, 2015).

The scale of modeling tsunami inundation inland with an explicitly represented constructed environment lies between that of modeling the large-scale tsunami wave propagation offshore and the small-scale tsunami impact on individual structures. This process is actually even more challenging to model since for two-dimensional depth-integrated models, inclusion of the constructed environment increases the complexity of the topography and the flow begins to have more variation in the vertical direction, while for the three-dimensional model that solves the Navier-Stokes equations, a fine mesh needs to be generated around each individual structure, which dramatically increases the number of cells in the computational domain.

Some researchers have tried to model this process with two-dimensional models. Ozer Sozdinler et al. (2015) used the numerical code NAMI DANCE to investigate hydrodynamic parameters in tsunami inundation zones with idealized structures – three rows of 20 blocks representing three-story concrete buildings. The code solved the NSWE using a finite-difference technique in a staggered leapfrog scheme. The effect of wave period, wave shape, protection structures, building layout and Manning's friction coefficient are discussed. Some major conclusions included that the coastal protection structures like seawalls and breakwaters have very limited effect if the waves are able to overtop them and that it is preferable to use different Manning's coefficients for the sea, land and buildings if more accurate values of hydrodynamic parameters are needed, but at the expense of more computational time. Similar conclusions on the Manning's coefficient were presented by Park et al. (2013). They simulated tsunami inundation in part of Seaside, Oregon and compared flow parameters with their physical experiment. The comparison showed that the flow parameters were sensitive to the friction coefficient, especially for the momentum flux, which is proportional to tsunami loads on structures. For instance, decreasing the friction coefficient by a factor of 10 increased the predicted momentum flux by 208%. Muhari et al. (2011) compared three different tsunami inundation models for evaluating tsunami impact on coastal communities: 1) a Constant Roughness Model (CRM) which uses a constant friction coefficient and does not include the constructed environment and assumes that all buildings are not able to withstand the tsunami; 2) a Topographic Model (TM) which includes the constructed environment by incorporating building shape and height information into the topography; 3) an Equivalent Roughness Model (ERM) which represents the building by using a different equivalent friction coefficient at the site of a building on the original topography (with only terrain information but not building height). Both the TM model and the ERM model gave more reliable prediction than the CRM model did, which confirmed the importance of taking the constructed environment into consideration.



However, few researchers have tried to use a three-dimensional model for inundation in a complex build environment. Shin et al. (2012) applied 3D LES (Large Eddy Simulation) model with two-phase flow to simulate inland tsunami inundation in a coastal city with hundreds of buildings and compared the prediction with experimental measurements. However, a fairly coarse mesh was used on land and each building had only 3 to 5 mesh cells along its edge in the along-shore or cross-shore direction, so that the resulting agreement in flooding depth can only be considered qualitative.

In this paper, the two models are first validated against an experiment in which a single bore impinges on a single column. Then they were used to simulate tsunami inundation of Seaside, Oregon, as represented by a physical model and experiments conducted by Park et al. (2013).

## 2 Simulation Methodology

### 2.1 Two Dimensional Model

The nonlinear shallow water equations can be written as

$$h_t + (uh)_x + (vh)_y = 0 \tag{1}$$

$$(hu)_t + (huv)_y + (hu^2 + \frac{1}{2}gh^2)_x = -ghB_x - Du \tag{2}$$

$$(hv)_t + (huv)_x + (hv^2 + \frac{1}{2}gh^2)_y = -ghB_y - Dv \tag{3}$$

where $u(x,y,t)$ and $v(x,y,t)$ are the depth-averaged velocities in the two horizontal directions, $h$ is the water depth, $g$ is gravitational acceleration, $B(x,y)$ is the topography, and $D = D(h,u,v)$ is the drag coefficient. The drag coefficient $D$ could have many forms; in this study it is represented by

$$D = \frac{gM^2\sqrt{(u^2 + v^2)}}{h^{5/3}} \tag{4}$$

where $M$ is the Manning's friction coefficient and is set to 0.025 for all two-dimensional simulations in this study. This value for the Manning's coefficient is the same as that used in the Constant Roughness Model of Muhari et al. (2011). The subscripts in these equations represent first order partial derivatives.

The GeoClaw model (LeVeque et al., 2011; Berger et al., 2011) features adaptive mesh refinement (AMR) and is released as a submodule of the Clawpack software (Clawpack Development Team, 2015), an open source package for solving hyperbolic systems of partial differential equations (PDEs) of one, two and three dimensions, through finite volume implementation of high-resolution Godunov-type "wave-propagation algorithms". Cell averages of the solution variables $q$ are computed over the volume of each cell and updated with waves propagating into the cell from all surrounding cell edges. The wave at each edge





is computed by solving a "Riemann problem" with initial piecewise constant data determined by cell averages on each side of the edge. This method is especially good at solving problems with discontinuous solutions like shock waves, which usually arise in the solution of nonlinear hyperbolic equations (e.g. bores in the case of NSWE).

Specifically, GeoClaw uses a variant of the $f$-wave formulation of the "wave-propagation algorithms" that allow incorpo-
ration of the topography source terms on the right hand side of equations 2 and 3 into the Riemann problem directly. The augmented Riemann solver in GeoClaw combines the desirable qualities of the Roe solver (Roe, 1981), HLLE-type (Harten, Lax, van Leer and Einfeldt) solvers (Einfeldt, 1988; Einfeldt et al., 1991) and the $f$-wave approach (Bale et al., 2003). The Roe solver provides an exact solution for the single-shock Riemann problem. It is also depth positive semidefinite like the HLLE solves, has a natural entropy-fix by providing more than two waves and yields a better approximation for problems with large
rarefactions. A large class of steady states is also preserved, even for non-stationary steady states with non-zero fluid velocity. In addition, it is able to handle the presence of dry states in the "Riemann problem", in which one state is wet ($h > 0$) while another is dry ($h = 0$), or both states are dry. It also works robustly in situations where the topography changes abruptly from one cell to another by an arbitrarily large value. For more details of the augmented Riemann solver in GeoClaw, see George (2008).

A typical characteristic of tsunami inundation models, especially those that incorporate the built environment, is that the spatial scale of regions of interest may vary from kilometers to meters. For regions several kilometers offshore, grid cells can be as large as thousands of meters on a side, while for regions near the shoreline or in a built environment onshore, grid cells must be refined to several meters or less, since the size of a building may be only several meters and an adequate number of grid cells are required to achieve acceptable accuracy. In GeoClaw, a patch-based AMR technique can efficiently handle these
situations (LeVeque et al., 2011; Berger and Leveque, 1998).

## 2.2   Three Dimensional Model

For the three-dimensional model, version 2.3.1 of the open-source CFD package OpenFOAM was used (The OpenFOAM Foundation, 2014). The package comes with different solvers for different types of flow. For tsunami inundation, in which there are two immiscible fluids (air and water) with a free interface, the interFoam solver can be chosen which uses the PISO
algorithm to solve the RANS equations with a volume-of-fluid (VOF) approach to model the free surface. For details of these numerical methods, readers can refer to Hirt and Nichols (1981); Versteeg and Malalasekera (2007). The VOF approach defines a scalar field $\alpha_{water}$ which represents fractional volume of water in each cell. A cell full of water ($\rho = 1000$ kg/m$^3$, $\nu = 1.0 \times 10^{-6}$ m$^2$/s) has $\alpha_{water} = 1.0$, while a cell full of air ($\rho = 1.22$ kg/m$^3$, $\nu = 1.48 \times 10^{-5}$ m$^2$/s) has $\alpha_{water} = 0.0$. Here $\rho$ is the mass density of the fluid and $\nu$ is the kinematic viscosity. A cell with $\alpha_{water}$ between 0 and 1 contains the interface.
A special transport equation is solved to advance the $\alpha_{water}$ field. To close the RANS equations, Menter's $k$-$\omega$-SST model (Menter and Esch, 2001) was applied.

There are many other turbulence closure models, among which the $k - \epsilon$ model is also very popular. It is suitable for fully turbulent and non-separated flows and has the shortcoming of numerical stiffness in the viscous sublayer, which can result in stability issues (Menter, 1993). It was also applied to model the inundation process in this study but became unstable during





the simulation. The $k$-$\omega$-SST is generally more stable and behaves better in modeling partially separated flows, which is the case in the current study (flow becomes separated after passing around the built environment).

With the assumption of an incompressible fluid, the RANS equations are listed below:

$$\frac{\partial \overline{u}_i}{\partial x_i} = 0 \tag{5}$$

$$\rho \frac{\partial \overline{u}_i}{\partial t} + \rho \overline{u}_j \frac{\partial \overline{u}_i}{\partial x_j} = -\frac{\partial \overline{p}}{\partial x_i} + \mu \frac{\partial^2 \overline{u}_i}{\partial x_j \partial x_j} - \frac{\partial \rho \overline{u_i' u_j'}}{\partial x_j} \tag{6}$$

where $\overline{u}_i$ is the mean velocity in the $i$ direction, $u_i'$ is the fluctuating component of velocity in the $i$ direction and $\overline{p}$ is the mean pressure. If $u_i$ is the velocity component in the $i$ direction, then $u_i = \overline{u}_i + u_i'$. The Reynolds Stress term in equation (6) is:

$$-\rho \overline{u_i' u_j'} = \nu_t \rho \left[ \frac{\partial \overline{u}_i}{\partial x_j} + \frac{\partial \overline{u}_j}{\partial x_i} \right] - \frac{2}{3} k \rho \delta_{ij} \tag{7}$$

where $k$ is the turbulence kinetic energy and $\nu_t$ is the turbulence eddy viscosity. The equations above need to be closed with some closure model. Here Menter's $k$-$\omega$-SST model (Menter and Esch, 2001) was applied:

$$\frac{\partial k}{\partial t} + \nabla \cdot (\mathbf{U}k) = \widetilde{G} - \beta^* k \omega + \nabla \cdot [(\nu + \alpha_k \nu_t) \nabla k] \tag{8}$$

$$\frac{\partial \omega}{\partial t} + \nabla \cdot (\mathbf{U}\omega) = \gamma S^2 - \beta \omega^2 + \nabla \cdot [(\nu + \alpha_\omega \nu_t) \nabla \omega] + (1 - F_1) CD_{k\omega} \tag{9}$$

where $\nu$ is the kinematic viscosity of fluid and $\widetilde{G}$ is defined as $\widetilde{G} = \min\{G, c_1 \beta^* k \omega\}$, where $G$ is the production term and defined as:

$$G = \nu_t S^2 \tag{10}$$

and $S$ is the invariant measure of the strain rate, defined by:

$$S = \sqrt{2 S_{ij} S_{ij}} \tag{11}$$

and $S_{ij}$ is the strain rate tensor defined by $S_{ij} = \frac{1}{2} \left( \nabla \mathbf{U} + \mathbf{U}^T \right)$. $F_1$ is a blending function defined by:

$$F_1 = \tanh \left\{ \left\{ \min \left[ \max \left( \frac{\sqrt{k}}{\beta^* \omega y}, \frac{500 \nu}{y^2 \omega} \right), \frac{4 \alpha_{\omega 2} k}{CD_{k\omega}^* y^2} \right] \right\}^4 \right\} \tag{12}$$

where $CD_{k\omega}^*$ is defined by:

$$CD_{k\omega}^* = \max \left( CD_{k\omega}, 10^{-10} \right) \tag{13}$$





and $CD_{k\omega}$ is defined by:

$$CD_{k\omega} = 2\sigma_{\omega 2}\nabla k \cdot \frac{\nabla\omega}{\omega} \tag{14}$$

After solving equations (8) and (9), $\nu_t$ can be calculated by:

$$\nu_t = \frac{a_1 k}{\max(a_1\omega, SF_2)} \tag{15}$$

where $F_2$ is a second blending function defined as:

$$F_2 = \tanh\left\{\left[\max\left(\frac{2\sqrt{k}}{\beta^*\omega y}, \frac{500\nu}{y^2\omega}\right)\right]^2\right\} \tag{16}$$

All other constants are computed using a blend from the corresponding constants associated with the $k$-$\epsilon$ and $k$-$\omega$ models via blending functions like $\phi = \phi_1 F_1 + \phi_2(1 - F_1)$. Values for these constants are: $\alpha_{k1} = 0.85013, \alpha_{k2} = 1.0, \alpha_{\omega 1} = 0.5, \alpha_{\omega 2} = 0.85616, \beta_1 = 0.075, \beta_2 = 0.0828, \gamma_1 = 0.5532, \gamma_2 = 0.4403, \beta^* = 0.09, a_1 = 0.31, c_1 = 10.0$ (Menter et al., 2003).

A force vector, $\mathbf{F}$, on a structure is computed by summing forces from pressure, $\mathbf{F}_p$, and from viscous stress , $\mathbf{F}_v$.

$$\mathbf{F} = \mathbf{F}_p + \mathbf{F}_v \tag{17}$$

$\mathbf{F}_p$ and $\mathbf{F}_v$ are calculated respectively by:

$$\mathbf{F}_p = \sum_i \left(-p_i A_i (\alpha_{water})_i \mathbf{n_i}\right) \tag{18}$$

$$\mathbf{F}_v = \sum_i \left\{(\tau_\mathbf{i} \cdot \mathbf{n_i}) A_i (\alpha_{water})_i\right\} \tag{19}$$

where $i$ is the index of cell faces on the building on which forces need to be evaluated, $p_i$ is the total pressure on face $i$, $A_i$ is area of face $i$, $(\alpha_{water})_i$ is volume fraction of water in the adjacent cell of face $i$, $\mathbf{n_i}$ is the unit normal vector of face $i$ pointing into the computational domain and $\tau_\mathbf{i}$ is the viscous stress tensor at face $i$ which can be expressed by $\tau_\mathbf{i} = \left\{\rho(\nu + \nu_t)\left[\nabla\mathbf{U} + \nabla\mathbf{U}^T\right]\right\}$ on face $i$.

## 3  Initial Comparison of The 2D and 3D Numerical Models

An initial comparison of the two numerical models was conducted by modeling the interaction between a bore and a free-standing coastal structure, with experimental results from Árnason (2005). The experiment was performed at the Charles W. Harris Hydraulics Laboratory at the University of Washington (UW), Seattle. In the experiment, a square column was placed in a 16.6 m long, 0.6m wide and 0.45 m deep wave tank, and aligned in parallel to the tank side walls (Fig. 1).

A thin gate separated water in the tank into two parts with different depths: 0.02 m deep on the square column side and 0.25 m deep on the other side. When the gate was lifted to the top of the tank in 0.2 s by a 6.4-cm diameter pneumatic piston, a bore





formed and propagated toward the square column downstream. The square column with a $12 \times 12$ cm square-shaped cross section was placed $5.2$ m downstream from the gate. To measure hydrodynamic forces, the column was supported from above and connected with a force sensor.

Both the three-dimensional and two-dimensional models were developed at model scale to simulate the physical experiment.
The three-dimensional OpenFOAM model incorporated the column into the computational domain by simply cutting off a block of mesh of the same shape from the computational domain. The mesh was coarse far from the column (1 cm by 1 cm by $0.5$ cm in the $x$, $y$, $z$ directions where the $z$ direction is perpendicular to the flume bottom) and was refined gradually to $0.125$ cm by $0.125$ cm by $0.0625$ cm in the $x$, $y$, $z$ directions near the column surface. The mesh was finer in the $z$ directions to better capture the water surface. Forces on the column were obtained by integrating pressure and shear forces from fluid on
the surface of the column.

In the two-dimensional GeoClaw model, the column was incorporated into the computational domain through the topography term $B(x,y)$ on right hand side of equations 2 and 3. Values for $B(x,y)$ are set to a very large constant value, $h_c$, in the region of the column and to $0$ elsewhere. This prevents water from overtopping the area, thus simulating a column. Setting $h_c$ to a very large value also made all four side walls of the square column be more "vertical" in the model since they are represented by
steep slopes arising from $B = 0$ (outside the column) to $B = h_c$ (inside the column). The coarsest level grid had a resolution of $0.02$ m by $0.02$ m and covered most of the computational domain; the finest mesh near the column was $0.25$ cm by $0.25$ cm.

First, a case without the column was modeled. Fig. 2 shows predictions of water level history, measured at $5.2$ m downstream from the gate (i.e., at $x = 11.1$, the center of the column. See Fig. 1 for location of the gauge) by the two numerical models and the experiment. In general, both 2D and 3D models accurately predict the arrival time of the bore, which is $t = 3.2$ s.
The OpenFOAM model matches the measurement better than GeoClaw with a sharp (but not vertical) slope at the front, a gradually rising surface to the peak near $t = 8$ s, then a downward slope, followed by interactions with the reflected wave from the back wall that creates the second jump in water level at around t = 14 s.

OpenFOAM includes water viscosity, which diffuses sharp discontinuities. In contrast, solving the nonlinear shallow water equations with an initial discontinuity yields a shock wave (discontinuity) propagating to the right as a vertical bore front
followed by a region with constant water depth; as a consequence, GeoClaw slightly overestimates the initial height of the bore front, underestimates the height at t= 8 s, and presents the reflected wave as a second sharp discontinuity at t = 13.1 s.

At the same location, streamwise (the along-channel direction) components of the velocity at different depths were also predicted. Fig. 3 shows time histories of streamwise velocity at 9 different distances from the bottom. Note that since the two-dimensional model is depth-averaged, its predicted velocity is constant with depth. The prediction from the two-dimensional
model matches the measurements very well near the water surface, except for the spike at the front, which is better captured by the three-dimensional model. The three-dimensional model underestimates flow velocity near the bottom might be due to our near-wall treatment is not perfect. The velocities at upper region are also hard to predict well because of air entrained in the water near free surface as well as the fact that the velocimeter may not be immersed in water at times, causing the measurement oscillates dramatically.



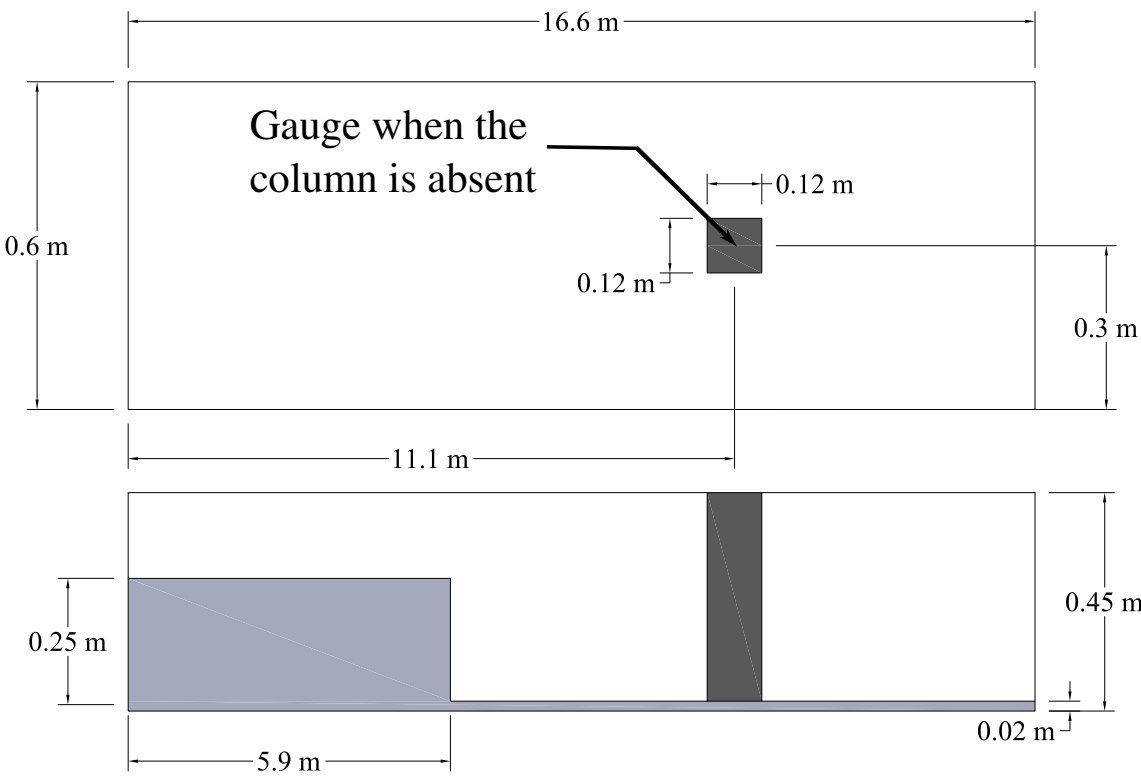

**Figure 1.** Schematic of the experimental setup for the interaction between bore and square column. The top figure shows a plan view and the bottom figure shows a cross section through the center of the column, illustrating also the bore.(Reprinted with permission from Motley et al. (2015). Copyright by ASCE.)







**Figure 2.** Time history of water level at 5.2 m from the gate (center of the column) with the column removed



**Figure 3.** Time history of streamwise velocity at different distances, $d$, from the bottom at 5.2 m from the gate (center of the column) with the column removed. Abscissa: time ($s$). Ordinate: velocity ($m/s$).





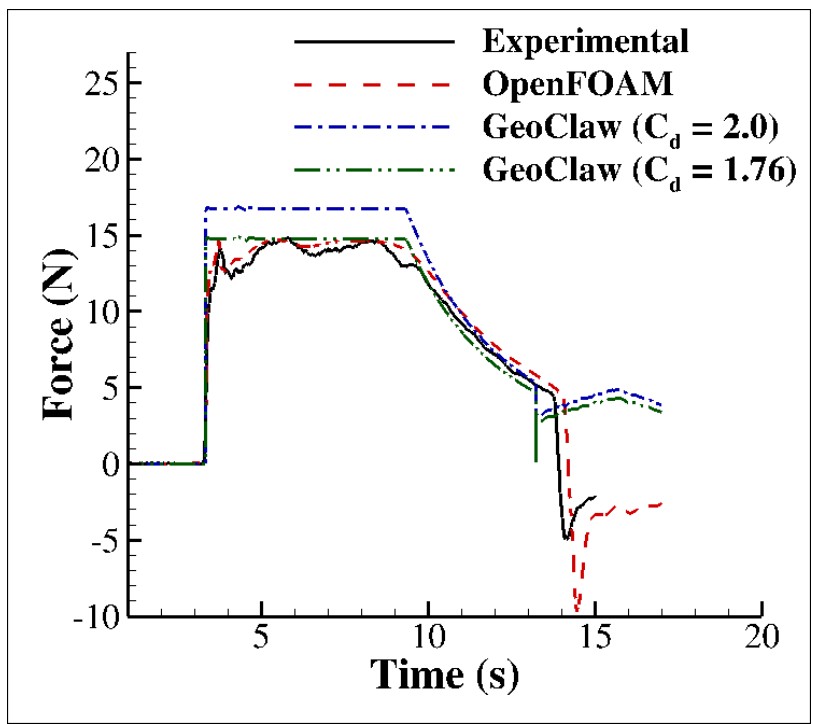

**Figure 4.** Comparison of measured and predicted horizontal forces on the square column. Sampling frequency is 300 Hz in the experiment and 1000 Hz in both numerical models.

Fig. 4 shows a comparison of total forces on the square column from the experiment, the three-dimensional model and the two-dimensional model. The force predicted by the three-dimensional model was obtained by integrating the pressure and viscous fluid forces on the surface of the column (See Eq. 17). The three-dimensional model predicts the force very well in terms of magnitude and is able to capture even the small spike near $t = 4$ s. In the two-dimensional model, no hydrodynamic

5   pressure field is available for force prediction. To predict forces from the two-dimensional model, data from the previous case without the column was used instead. The water level, $h$, and streamwise velocity, $u$, were first sampled at the center of the footprint of the column that was removed from the domain, to compute the momentum flux, $M = hu^2$. As recommended by FEMA P-646 (2012), the hydrodynamic forces on such a structure can be computed as

$$F_d = \frac{1}{2} C_d \rho (hu^2) b \qquad (20)$$

10   where $C_d$ is the drag coefficient and may be conservatively chosen to be $2.0$ as recommended by FEMA P-646 (2012), $F_d$ is the streamwise component of the fluid forces, $\rho$ is the density of the fluids, $h$ is the water depth, $u$ is the fluid velocity at the location of the structure, and $b$ is the breadth of the structure in the plane normal to the direction of flow. Note that the $hu^2$ term in the denominator is the momentum flux, $M$.





Note that in the experiment or three-dimensional model, the water level on the upstream side of the column is different from that on the downstream side of the column. This causes a difference in hydrostatic pressure and thus a hydrostatic force on the column. For this reason, it may be more appropriate to refer to this value as the coefficient of resistance instead of solely as a drag coefficient. Using a drag coefficient of 2.0 overestimates the force by 13% in general. This is as expected since it is said

to be "conservative" according to FEMA P-646 (2012). Fig. 4 also shows that if a drag coefficient of 1.76 is used instead, the force prediction from the two-dimensional model matches the measurement more closely.

## 4   The Seaside Wavetank Model

### 4.1   The Physical Experiment

A 1:50 scale physical model of part of Seaside, Oregon, adjacent to the Cascadia Subduction Zone (CSZ), was constructed in the

Tsunami Wave Basin at the O.H. Hinsdale Wave Research Laboratory at Oregon State University, and a series of experiments were conducted to measure flow velocities and water levels at 31 locations within the model-scale community. For full details of the experiment, one can refer to Park et al. (2013).

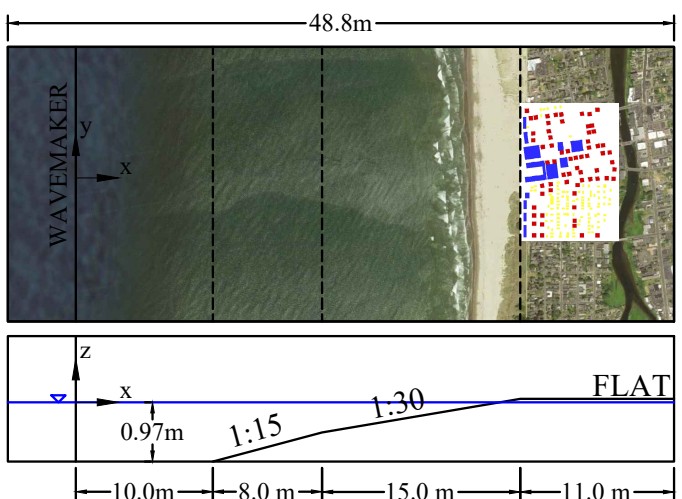

**Figure 5.** Top view and side view of the basin.

The rectangular basin for the experiment is 48.8 m long, 26.5 m wide and 2.1 m deep. Fig. 5 shows the top and side view of the basin. The still water depth at the wavemaker is 0.97 m and decreases as it approaches the shoreline. A 0.04 m height

(model scale) seawall was also constructed between all idealized buildings and the shoreline and was parallel to the wave maker. Figs. 6 and 7 show the locations of the 31 gauges where water level and flow velocity were measured at a sampling



frequency of 50 Hz in the experiment. The gauges are grouped into 4 groups, A, B, C and D (from bottom to top) and marked by different symbols. Buildings in blue are large commercial buildings like hotels and hospitals. All red buildings are of the same size and represent small commercial buildings. Buildings in yellow are residential structures and are also all the same size.

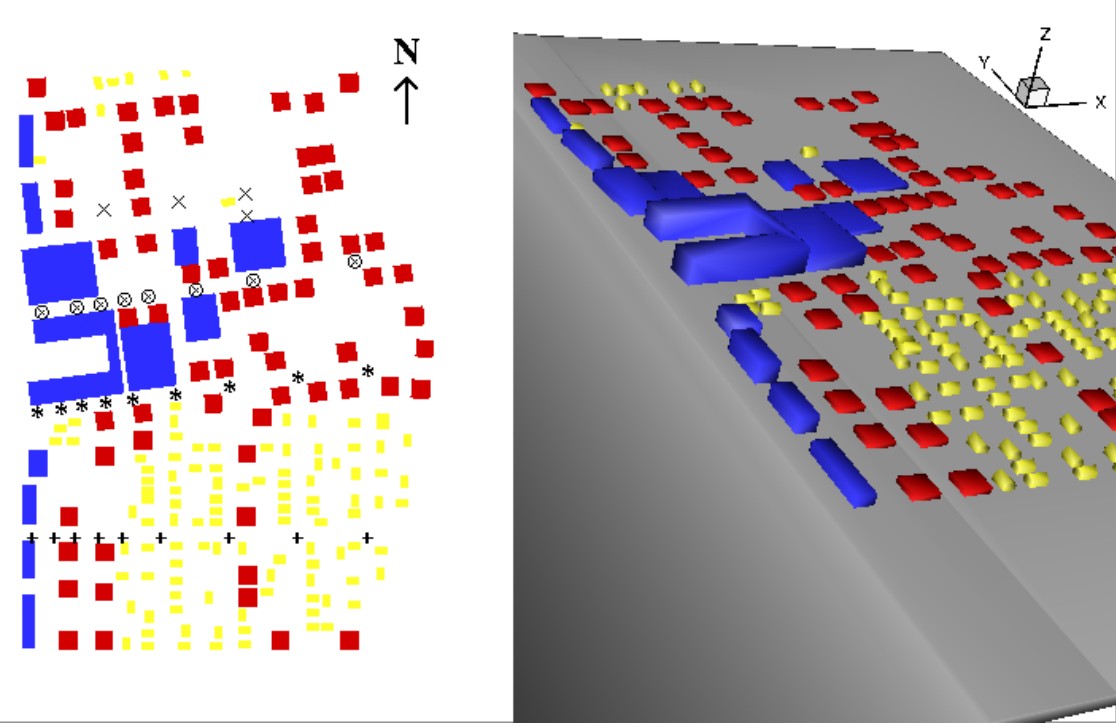

**Figure 6.** Layout of all buildings and gauges in the experiment: blue, large hotels or commercial buildings, red, smaller commercial buildings, yellow, residential structures. $+$: gauge A1 $\sim$A9; $*$: gauge B1$\sim$B9; $\otimes$: gauge C1$\sim$C9; $\times$: gauge D1$\sim$D4.

5    In the experiment, the piston-type wave maker was designed to generate an initial wave with a wave height of approximately 0.2 m (model scale) at the lower horizontal section of the basin; this is equivalent to 10 m at full scale, which corresponds to a 500-year CSZ tsunami for this region (Tsunami Pilot Study Working Group, 2006). Note that this is not a solitary wave but a long single-peak wave. Experimental measurement of the wavemaker speed was fit with a Gaussian function of the form $s(t) = Ae^{(\beta(t-t_0)^2)}$, which was used as input to generate numerical wave in current simulation. The experiment was repeated

10   many times with identical initial conditions. Data from multiple trials were averaged to obtain the results presented here to smooth out stochastic features of the experiment, more details of which were presented in Park et al. (2013).





## 4.2 Setup of Numerical Models

### 4.2.1 OpenFOAM Model

In the three dimensional OpenFOAM model, a numerical wave basin was developed to simulate the experiments. It was built at the model scale instead of full scale to exclude scaling effects. This facilitated the comparison between the numerical model
and the physical experiment.

To generate the required waves, a numerical wave generator was previously developed in OpenFOAM (Motley et al., 2014) and it was validated against available data from a pair of experiments. Two steps are taken by the numerical wave generator to simulate wave generating procedure of a piston-type wave maker. First, a short subsection of the wave basin adjacent to the wave maker is modeled. This step is conducted with the wave maker as the reference frame, eliminating the need for a
moving mesh, and fluid is forced to enter the domain at the wave maker's speed from the other end of the domain to simulate the movement of the wave maker. A time-varying acceleration vector field is also embedded in the solver to compensate for the non-inertial frame. The second step is to map all field data in this domain (the generated wave) to a full model of the basin with the mapFields utility in OpenFOAM, after the wave maker stops moving. Further simulations can then start from here.

One disadvantage of the three dimensional model is that it requires heavy computational resources. Even with 4 dual 8-core
2-GHz Intel Xeon e5-2650 machines (64 total processors), it was not possible to model the entire basin. Instead, the entire domain was divided into four different subsections of equal width to predict flow parameters at different groups of gauges (See Fig. 7). For clarity, only the onshore domain is shown in the figure; however, the numerical domain spans the entire 48.8 m from the wavemaker to the back wall of the basin. For each simulation, approximately 60 million cells were used and the solver was run in parallel with 64 processors cores mentioned above for ~10 days (including wave generation), which is equivalent to
a total CPU time of ~640 days.

The boundary conditions for each boundary in the numerical wave basin are listed in Table 1. The term *All walls and floor* in the table includes the bottom, side walls, two end walls and surfaces of internal buildings. Another term, *Atmosphere*, refers to the upper boundary of the computational domain. A *zeroGradient* boundary condition specifies zero gradient on the boundary. A *fixedValue* boundary condition sets the value of a quantity to a constant specified value on the boundary. The velocity field
on a wall is set to 0. An *inletOutlet* boundary condition is identical to the *zeroGradient* boundary condition if the flux is out of domain but is switched to apply a *fixedValue* boundary condition if the flux is into the domain. The *pressureInletOutletVelocity* condition at the top of the domain is essentially identical to a *zeroGradient* boundary condition in our current model. On *All walls and floor*, $p_{rgh}$ is defined such that there is zero flux, using the *fixedFluxPressure* boundary condition, while the *Atmosphere* was defined with a uniform reference pressure $p_0$ using the *totalPressure* boundary condition:

$$p_{rgh} = \begin{cases} p_0 & \text{, for outflow} \\ p_0 - \frac{1}{2}|\mathbf{U}|^2 & \text{, for inflow} \end{cases} \tag{21}$$



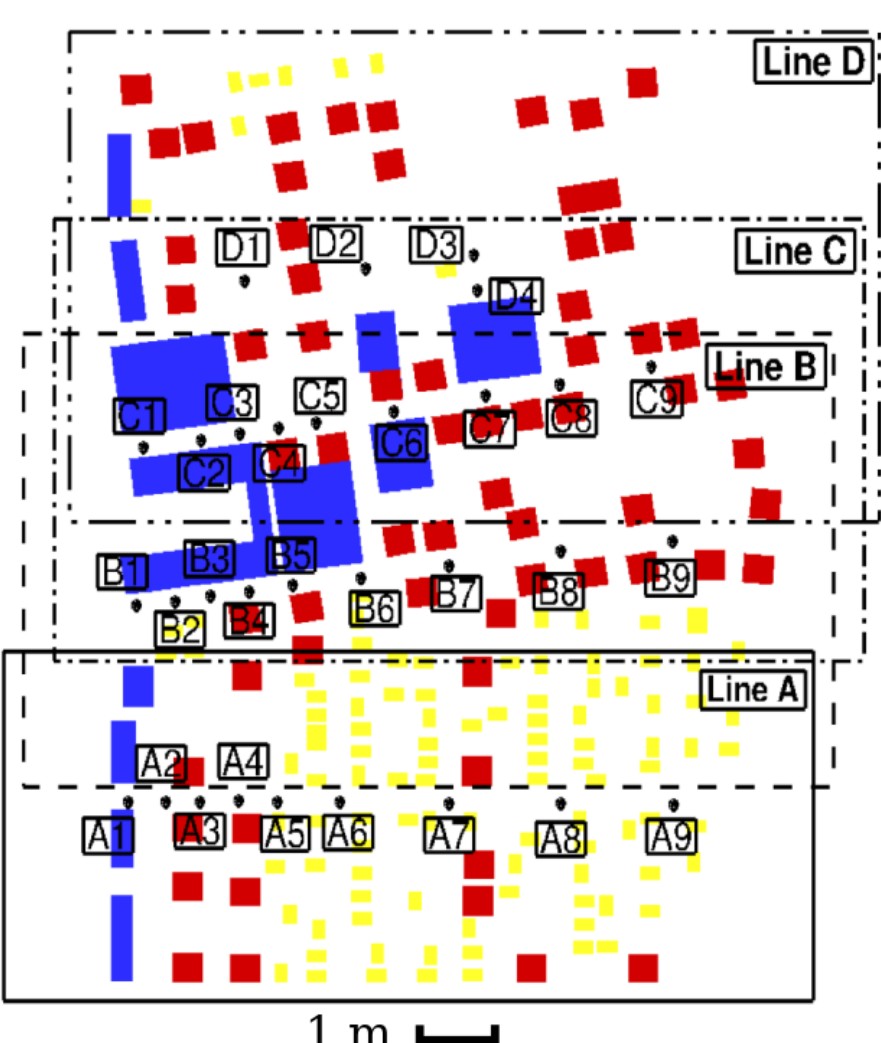

**Figure 7.** Four different subsections and layout of gauges





**Table 1.** OpenFOAM boundary conditions for the current numerical model

| Field | All walls and floor | Atmosphere |
| --- | --- | --- |
| Air/water phase indicator, $\alpha_{water}$ | zeroGradient | inletOutlet |
| Velocity, $\mathbf{U}$ | fixedValue | pressureInletOutletVelocity |
| Pressure without hydrostatic part, $p_{rgh}$ | fixedFluxPressure | totalPressure |
| Turbulent kinetic energy, $k$ | kqRWallFunction | inletOutlet |
| Specific dissipation rate, $\omega$ | omegaWallFunction | inletOutlet |
| Turbulence eddy viscosity, $\nu_t$ | nutUSpaldingWallFunction | zeroGradient |

Here $p_{rgh}$ is pressure subtracted by static pressure $\rho g h$ where $\rho$ is the water density, $g$ is the gravitational acceleration and $h$ is relative depth under initial free surface. The turbulence quantities near solid walls are obtained with wall functions that model them as functions of distance from the boundary.

Centers of the first layer of cells near the wall are chosen as positions in the log-law region of the boundary layer where the
wall functions are applied. A *kqRWallFunction* boundary condition can be expressed as $\frac{\partial k}{\partial n} = 0$ for $k$ on a wall where $n$ is a unit normal vector to the wall. An *omegaWallFunction* boundary condition provides a wall function for the turbulence specific dissipation, $\omega$, with default model coefficients: $E = 9.8$, $\kappa = 0.41$, $C_\mu = 0.09$. It is computed with:

$$\omega = \sqrt{\omega_{vis}^2 + \omega_{log}^2} \qquad (22)$$

where $\omega_{vis}$ is the value of $\omega$ in the viscous region and $\omega_{log}$ is the value of $\omega$ in the logarithmic region (Menter and Esch, 2001).
The *nutUSpaldingWallFunction* boundary condition for $\nu_t$ is used for smooth walls. It computes a continuous $\nu_t$ profile to the wall based on Spalding's law (Spalding, 1961), which is essentially a unified law of the wall which works for the viscous sublayer, buffer layer and the logarithmic region in a boundary layer.

The initial condition for $\alpha_{water}$ is set to 1 for cells where there is water at the beginning and to 0 for the rest. The initial value of $\mathbf{U}$ and $p_{rgh}$ were zero since the flow is initially at rest. Although the fluid is at rest at the beginning, a small value of
the turbulent kinetic energy $k$ must be "seeded" in the domain, because the production term in the governing equation of the turbulent kinetic energy $k$ is zero and thus will produce no turbulence if initially $k$ is zero.

Assuming zero velocity fluctuation in the along-shore and vertical direction, the definition of $k$ gives:

$$k = \frac{1}{2}(u_1'^2 + u_2'^2 + u_3'^2) \approx \frac{1}{2}u_1'^2 \qquad (23)$$

The velocity fluctuation $u_1'$ is computed from $I = \frac{u'}{U}$ where $I$ is the turbulence intensity, $u' = \sqrt{\frac{1}{3}(u_1'^2 + u_2'^2 + u_3'^2)}$ and $U$ can
be chosen as wave celerity in this case. This approach is the same as Svendsen (1987) and Lin and Liu (1998). Several choices of initial turbulence intensity was tested. To best match the wave height at wave gauge WG1 and WG3, an initial turbulence





intensity of $1\%$ is chosen in this model. For the specific dissipation rate, $\omega$, $\omega = \frac{\sqrt{k}}{l}$ is used where $l$ is the turbulent length scale and is set to $7\%$ of the hydraulic diameter of the channel-like computational domain, according to Pope (2001).

Two computational meshes with refinement focused on different regions were used during the simulation to mitigate computational demand. The first mesh was used in the first phase, from the beginning of the simulation to the time when the wave almost started to break. In this mesh, all buildings were removed from the domain, leaving only the flat bottom of the wave basin, which reduced the number of cells needed onshore significantly, allowing for use of a much finer mesh offshore. The mesh size was approximately 0.08 m × 0.08 m × 0.01 m (length × width × height) near the wave maker and was gradually reduced to 0.08 m × 0.08 m × 0.004 m onshore due to changes in topography. Note that the mesh cells onshore seem to have large aspect ratio but there is no water onshore at all during this time period. The second mesh was used until the end of simulation. The buildings were added into the domain and very fine mesh was generated around the the onshore bathymetry. The mesh size was 0.3 m × 0.015 m × 0.035 m near the wave maker and refined to 0.0075 m × 0.0075 m × 0.0025 m near the flat bottom of the onshore segment of the basin and at the edges and corners of the buildings. It should be noted that this was only the size of a structured background mesh, which was further refined by a factor of 2 and deformed by a mesh tool, snappyHexMesh, from OpenFOAM near the buildings and seawalls to make the mesh accurately represent the complex and irregular geometry of the boundaries. Simulation results from the end of the first phase were mapped to the second phase and the simulation continued. This strategy is similar to the dynamic adaptive mesh refinement (AMR) feature in the 2D GeoClaw model. Here, however, statically refined meshes were used instead of dynamically refined grids used in the 2D GeoClaw model. The average Courant number across the entire computational domain during these simulations is approximately 0.01. While this is considerably low for a typical analysis, this is due to the fact that grid sizes vary by several orders of magnitude.

### 4.2.2 GeoClaw Model

With GeoClaw, it is possible to model the entire basin. Thus, the computational domain is a 48.8 m by 26.5 m rectangle. The geometry of the basin bottom and built environment are described by topography files of different resolution, which specify $B(x,y)$ on the right hand side of equations 2 and 3. A typical wall time for one simulation is approximately six hours with a single core in an Intel(R) Core(TM) i7-4790 CPU processor, which means the CPU time is also six hours (0.25 day). Note that the computational resources required by the GeoClaw model is only $0.25 \div 640 \approx \frac{1}{2500}$ of what is required by the three-dimensional OpenFOAM model in this study.

To generate tsunami waves in GeoClaw, user defined time varying boundary conditions can be specified at the inlet of the computational domain, based on data for the wavemaker speed $s(t)$ in the physical experiment. The data from the physical experiment can be fit quite well with a Gaussian of the form

$$s(t) = Ae^{\beta(t-t_0)^2} \tag{24}$$

with $\beta = 0.25$, $t_0 = 14.75$ and amplitude $A = 0.51$. However, the way we imposed velocity boundary conditions at a fixed location rather than having a moving boundary, we found better agreement with the observed wave at several offshore wave gauges by setting A = 0.6 in equation 24, which was therefore used for all simulations.





The AMR feature of GeoClaw was used, with a mesh size for the base-level grid of 0.5 m (corresponding to 25 m in full scale) in both cross-shore direction and along-shore direction. The term cross-shore is used to refer to the direction that the wave propagates from the wavemaker to the structures onshore, while the direction perpendicular to the cross-shore direction is referred to as the along-shore direction. The mesh is refined in the nearshore region up to 4 levels, with specified refinement

ratios: 4 from level 1 to 2, 5 from level 2 to 3 and 2 from level 3 to 4. The finest mesh in the domain with this setup for AMR is 0.0125 m by 0.0125 m (corresponding to 0.625 m in full scale) and eventually covers the entire onshore region. The desired Courant number is set to 0.9 to guarantee the stability of the explicit numerical scheme.

One thing to be noted is that for both numerical models described above, all coastal structures, including different types of buildings and the seawall, are assumed to be undamaged and thus fixed and rigid during the inundation.

**4.3   Comparison of Flow Parameters**

The predicted free surface elevation, cross-shore velocity, and corresponding momentum flux from the two numerical models will be compared and discussed in this section. All experimental data in this study were provided by the NTHMP Mapping and Modeling Benchmarking Workshop: Tsunami Currents (University of Southern California, 2015), and descriptions of the physical experiments to gather the data are provided by Park et al. (2013) and Rueben et al. (2011).

Gauges were positioned as shown in Figs.5-7. Ultra-sonic surface wave gauges (USWG) were used to measure the free surface. The bore front propagation speed was obtained by analysis of imagery gathered by two high resolution video cameras located above the wave basin (Rueben et al., 2011). Fluid velocity measurements were acquired by Acoustic Doppler Velocimeter (ADV) only after peaks; air entrainment in the bore at and shortly after the initial impact rendered the ADV measurements inconsistent in repeated trials (Park et al., 2013). Park et al. (2013) then assumed that the propagation speed and fluid velocity

at the bore front are equal and fit a second-order polynomial to that value and ensemble-averaged ADV measurements in this region.

Time histories of the free surface elevation, cross-shore velocity and corresponding momentum flux at selected gauges are shown in Figs. 8-11. After the peak (initial impact), there appears to be a significant drop in discrepancies between modeled and measured water level and fluid velocity; therefore, the discussion that follows will separately compare the results before

and after the peak.

**4.3.1   Onshore time series near initial impact**

Water level amplitude by OpenFOAM and arrival time by both OpenFOAM and GeoClaw agree fairly well with measurements at many of the gauges in groups A, B and C, but GeoClaw underestimates the amplitude at many gauges. These differences reflect the challenge of modeling a turbulent and rapidly varying bore front. An additional factor is that the gauges in groups

A, B and C are placed along straight lines, representing roads within the community, whereas those in group D are set behind buildings. As a consequence, flow around group A, B and C gauges is dominated by flow in the cross-shore direction, while flow around group D gauges is more complex and challenging to model.



Fluid velocity experimental values derived by optical means are significantly lower than the modeled OpenFOAM and GeoClaw velocity in many of the 16 cases presented in Figs. 8-11. This is because the optical measurement of the bore front is not necessarily representative of flow velocity. Here the animation of GeoClaw numerical results was analyzed to obtain estimates of 1.3m/s for peak velocity: Fig. 12 showed modeled velocity distributions in the bore at two consecutive time steps

in the GeoClaw simulation at gauge A4, illustrating that the modeled maximum fluid velocity occurs at some point behind the bore front.

Momentum flux modeled by OpenFOAM and GeoClaw do not agree well with experimental estimates, due to the discrepancies in fluid velocity estimates, discussed above. This is critical, since momentum flux is often used to compute the tsunami forces on structure, as discussed in detail in section 5.

In summary, predictions near the initial impact are challenging for both models, but the three-dimensional OpenFOAM model performs better than the two-dimensional GeoClaw model because it models turbulence and the variation of velocity with depth.

### 4.3.2 Onshore time series in post-impact region

Water level agreement among both models and the experimental data are significantly improved after initial impact. Note that

some gauges are quite far from the shoreline (for example, gauges A6, B8, C8), where the inundation depth is very shallow compared to the peak value near the shoreline (less than 20% of the peak value). Even at these locations, however, both numerical models provide reasonable predictions. It is also of interest that, as noted above, GeoClaw predicts a lower bore front propagation speed than OpenFOAM; as a result, arrival of the OpenFOAM bore front agrees well with experiment, but the GeoClaw bore front is significantly delayed at gauges farther inland, such as B8 and C8 (see Fig 9 and 10).

Fluid velocity measurements by the ADV are more stable after 30 s, and both OpenFOAM and GeoClaw velocity time series agree much better with the experimental data at gauges in groups A, B and C. Agreement does degrade significantly in group D, especially in the case of GeoClaw; this is no doubt due to the more complicated fluid flow in the group D environment, behind buildings, compared to the relatively simpler cross-shore flow in the street environments of groups A, B and C (Fig. 7).

Momentum flux from both numerical models are in better agreement with the measurements at most gauges, since water

level and velocity agreements are better than in the $t < 30s$ time period.

Fig. 13 compares snapshots of the simulation near line A from the two models at 3 different times. The three-dimensional model provides substantial detail about the complex flow among buildings, including the strong channeling effect along line A, aligned with the street, and among the buildings on both sides of the street. These channeling effects can alter the forces exerted on both sides of that street, so that any differences between OpenFOAM and GeoClaw in modeling such effects may

result in different prediction of forces on the buildings.

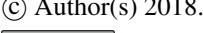





**Figure 8.** Time histories of surface elevation, cross-shore velocity and momentum flux at some selected gauges along line A (Note that ranges of Y axis are different in different subplots)





Gauge B1

Gauge B3

Gauge B6

Gauge B8

**Figure 9.** Time histories of surface elevation, cross-shore velocity and momentum flux at some selected gauges along line B





Gauge C1

Gauge C3

Gauge C6

Gauge C8

**Figure 10.** Time histories of surface elevation, cross-shore velocity and momentum flux at some selected gauges along line C







Gauge D1

Gauge D2

Gauge D3

Gauge D4

**Figure 11.** Time histories of surface elevation, cross-shore velocity and momentum flux at some selected gauges in group D




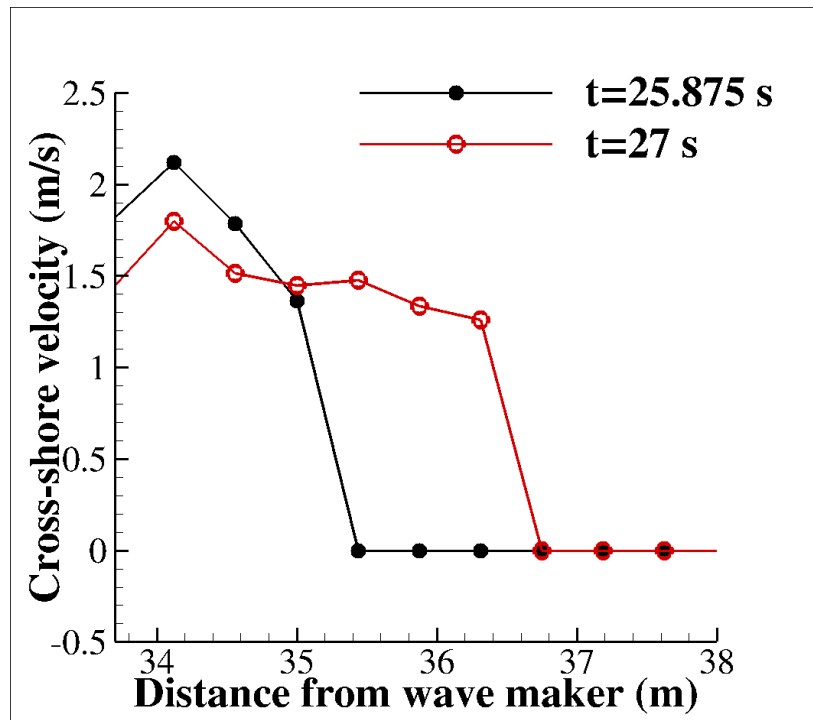

**Figure 12.** Velocity distribution in the bore near gauge A4, from the GeoClaw model

## 5   Force predictions from momentum flux

Some representative buildings along Line A were selected for preliminary analysis of fluid forces on the coastal infrastructure, as shown in Fig. 14. Building I is one of the two large structures adjacent to gauge A1 and directly facing the shoreline, with a dimension of 0.29 m by 0.78 m by 0.246 m (length in cross-shore direction by length in along-shore direction by height) and

0.31 m by 0.84 m by 0.31 m, respectively. Buildings III has a dimension of 0.39 m by 0.39 m by 0.091 m. Buildings III and IV, representing small houses within the community, are identical but placed in different directions, which has a length, width and height of 0.17 m, 0.26 and 0.154 m respectively.

In terms of force measurements, the single-column case presented in Section 3 was the only dataset available with experimental measurements of wave impact forces on similar structures. Through validation against that data, it was shown that,

provided the water height and fluid velocity are properly modeled, the fluid induced forces could also be properly predicted. This could be generally extrapolated and applied to the Seaside problem, where the only available measured data included flow parameters (water depth and velocity).

Fig. 15 shows predicted forces in the cross-shore direction from the two models on selected buildings. Note that these forces are normalized by the width of western (left) wall of the buildings. Since no pressure field exists in the two-dimensional

GeoClaw model, the same approach as was used in section 3 is applied here to compute forces on these selected buildings for





**Figure 13.** Snapshots of the simulation near line A, colored by cross-shore velocity, at 3 different times (from top to bottom): $t = 25.9$ s, $t = 27$ s, t=28.1 s. Left: Geoclaw; Right: OpenFOAM.





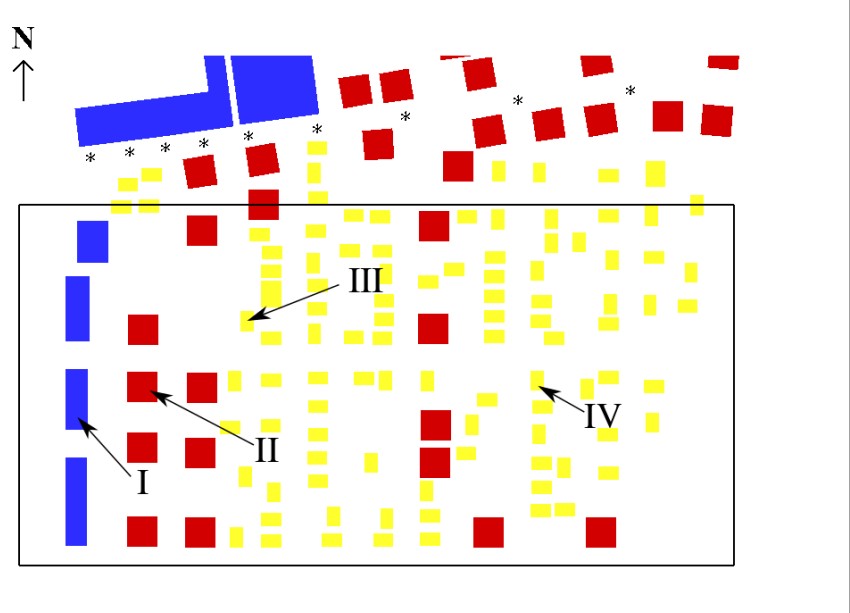

**Figure 14.** Representative buildings along Line A.

the GeoClaw model ($C_d$ chosen as 2.0 as well). In this case, note that not all the buildings are removed to get the momentum flux for a specific building. Instead, only the building at the center of which the momentum flux is to be predicted is removed with all other constructed environment unchanged. This minimizes the influence of removing that building on the flow overall.

Peak values of forces predicted by the GeoClaw model on all buildings are only approximately half of those predicted by

the OpenFOAM model, except for building III. This indicates that this approach for predicting tsunami load can be off by as large as 100% in such a complex scenario where multiple objects are present, although it is recommended by FEMA P-646 (2012) as an empirical method when only velocity and surface elevation map is available and is prevalent in tsunami inundation problems.

## 6   Conclusion and extensions

In this paper, two different types of numerical models of tsunami inundation were developed and compared. They were first validated by comparing water level, velocity profile and forces on a single column impacted by a bore from a dambreak. Then the two models were used to predict free surface elevation, velocity and momentum flux of a tsunami inundation on a model-scale constructed environment. The predicted flow parameters agree well with experimental measurements in the post-impact region at most gauges. During initial impact, however, the two-dimensional GeoClaw model has difficulty in capturing transient

characteristic of the flow. The three-Dimensional OpenFOAM model can solve this challenge better, but at the expense of much more computational resources required. This is because the variation in the vertical direction is "eliminated" by the integration





**Figure 15.** Predicted forces in cross-shore direction on selected buildings (normalized)




in two-dimensional model while all three-dimensional characteristics of the flow as well as turbulence are modeled by the three-dimensional model. Several primary conclusions can be drawn from this work:

1. The three-dimensional RANS model can predict flow parameters and forces on structures by modeling only a subsection of $\frac{1}{3}$ width of the entire basin, while the two-dimensional NSWE model can model the entire basin at one time, with much less computational resources. Both models agree well with experimental measurements at most locations considered after the initial impact. The RANS model, however, can provide more details of the flow, especially near the initial impact region.

2. The fluid dynamics in the bore front are transient and turbulent. Thus near the initial impact, prediction of flow parameters and forces is challenging but also the most critical since the flow parameters and forces have maximum value near this point. The three-dimensional RANS model solves this challenge better than the two-dimensional NSWE model but needs much more computational resources.

3. Using the approach recommended by FEMA P-646 to predict fluid forces on structures from the two-dimensional model works well in the simple case of flow around a column, but becomes less reliable in a complex constructed environment. Although choosing a drag coefficient of 2.0 is considered conservative, the 2D model with this value was still seen to significantly underestimate fluid forces (in some cases giving only half the correct value in the results discussed in Section 5) because the 2D model underestimates peak velocities in this complex flow.

This research compares different characteristics of a two-dimensional model and a three-dimensional model of tsunami inundation with constructed environment. Challenges in prediction of flow parameters and forces are revealed and the capabilities of the two numerical models in solving this type of problem are analyzed. A trade-off needs to be made between the two models due to their different levels of accuracy and required computational resources. The comparisons in the current study can provide a reference when choosing between two-dimensional model and three-dimensional model to predict required information in tsunami inundation.

*Competing interests.* None.

*Acknowledgements.* The authors would like to thank the National Science Foundation for their financial support through Grants EAR-1331412 and CMMI-1536198. This work was facilitated through the use of advanced computational, storage, and networking infrastructure provided by the Hyak supercomputer system.





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
