# Peer review of "A comparison of a two-dimensional depth averaged flow model and a three-dimensional RANS model for predicting tsunami inundation and fluid forces"

_Natural Hazards and Earth System Sciences, 2018_

## Referee Comment (RC1) · Anonymous Referee #1 · 27 Jun 2018

Summary of the Manuscript:

The authors present comparisons of simulations for tsunami inundation that incorporate the built environment of a physical model of Seaside, Oregon. The comparison has been made between a two-dimensional depth-integrated model namely GeoClaw and a three-dimensional model called OpenFOAM. Initially, the models were validated through a dam break experiment that compared the water level, velocity profile and forces on a single square column by a bore. Without the column, the OpenFOAM model was able to reproduce the water level fairly accurately whereas GeoClaw slightly over-

estimates the initial bore height followed by an underestimation as the bore progresses. Moreover, the vertical velocity profile captured by OpenFOAM was predicted well except at the bottom whereas the GeoClaw only gives constant velocity over depth which was determined to be overpredicted. With the column included OpenFOAM model captured well horizontal forces on a square column whereas the drag coefficient had to be decreased from 2.0 to 1.76 for the force prediction to match the measurement closely.

According to the authors, simulating tsunami inundation for a physical model of Seaside, Oregon which incorporated the built environment using the two models had their challenges especially in the case of the 3D model OpenFOAM. OpenFOAM is shown to be very computer intensive compared with GeoClaw. Both models were able to predict the flow well compared with the experimental model, however, OpenFOAM provided more details of the flow especially near the impact. Thus, according to the authors, the three-dimensional model solves this challenge better than the two dimensional one. The 2D model however, underestimates the forces as the model underestimates the flow velocity in the complex flow.

Overall Assessment:

Overall, I think the study makes a beneficial contribution in understanding the flow characteristics and forces acting in a complex constructed environment through the use of two and three dimensional models. The conclusion of the study can assist civil engineers in improving upon the designing of coastal structures in tsunami inundation zones as the 3D model gives a better representation of the forces. I would like to recommend this manuscript provided the author addresses the following comments.

Comments:

Page 1 - 5: I would suggest if the Introduction can be tailored to a broader audience and be more concise in terms of purpose, application and scope of the paper. Most of the existing introduction can then go under a section on Previous Work. Also the authors

could include examples (in possibly a separate section) of existing coastal structures in tsunami inundation areas that have utilised 2D or 3D modelling studies to determine forces on structures. You might want to cite:

Ingraffea, Nathan & Libby, Mark, 2015. Design of a Tsunami Vertical Evacuation Refuge Structure in Westport, Washington. Structures Congress 2015, pp.1530–1537.

González, Frank, Randy LeVeque, and Loyce Adams. "Tsunami Hazard Assessment of the Ocosta School Site in Westport, WA." (2013).

Page 1, Line 10: The line should read, "However, it is not clear whether these equations ..."

Page 3, Line 24: Has not the increased computing power affected both tsunami runup process and wave impact on an individual structure.

Page 8, Line 17: space after i,

Page 9, Line 33-34: the line should read, "..., causing the measurement to oscillate dramatically."

Page 13, Line 13: the momentum flux in equation 20 is in parenthesis so replace denominator by parenthesis

Page 15, Line 1: delete in the experiment. The sentence already makes it clear that the sampling rate is for the experiment.

Page 15, Line 7: Define CSZ here i.e. Cascadia Subduction Zone.

Page 28 - 30: The conclusion may be strengthened by suggestions for the practitioner as to when might it be useful to utilise three-dimensional model studies rather than two dimensional studies in designing coastal structures within tsunami inundation areas and whether the increased computational power is really necessary or not. This point may be connected to looking back on what may have been done differently when determining forces to design for example the Tsunami Vertical Evacuation Refuge Structure

in Westport.

Figure 3. Add legend, remove grid, add one label for time and velocity along x and y axis respectively so you can then remove Abscissa: time (s) a. Ordinates: velocity (m/s) from the caption.

---

## Referee Comment (RC2) · Anonymous Referee #2 · 28 Jun 2018

The paper compares tsunami inundation and forces based on two modelling approaches based on: (1) nonlinear shallow water equations, and (2) Reynolds-averaged Navier Stokes equations coupled with k-omega SST turbulence closure. The models are first validated against experiments involving bore impinging onto a single square column. They are then utilized to simulate tsunami inundation of a physical model of Seaside, Oregon, USA. Proper CFD simulations of tsunamis are relatively rare in the literature, hence making this work novel. Differences are found in details of the flow, e.g. near the initial impact, demonstrating the usefulness of CFD in this context. The

work is generally well organized and written, though I have several suggestions for improved clarity below. Overall, I suggest that the work be accepted pending minor revisions, wherein the more detailed comments below are addressed.

1. As mentioned above, CFD of tsunamis are relatively rare. As this is much of the novelty of the present work, a more thorough literature review on this general topic would seem warranted, as several seemingly relevant papers are not cited. Such works seemingly include: Biscarini (2010), Montagna et al. (2011), Larsen et al. (2017), as well as Aniel-Quiroga et al. (2018).

2. p. 5: B(x,y) is ambiguously defined as the topography. Does this mean the bed elevation? Please clarify.

3. p. 9: It is stated that z is perpendicular to the flume bottom. Would it not be simpler to state that z is vertical?

4. p. 9: Discussing mesh resolution strictly in dimensional terms gives little physical meaning. Please also add discussion in terms of wall units, $z^+=z*U_f/nu$, where $U_f$ is the friction velocity and nu the kinematic fluid viscosity.

5. p. 9: I am not convinced that simply making B(x,y) very large properly simulates a column. How exactly has this been tested? Why should the vertical column wall be modelled differently than other vertical walls?

6. Please add axes with labels to Figure 1, this will greatly improve clarity.

7. Forces are estimated using a drag coefficient in Eq. 20. Why is the more general Morrison equation not used?

8. p. 15: The function s(t) is given, but without specifying parameters A and beta, hence the reader is given no information regarding the duration. Please clearly define these parameters. Sufficient information must always be given such that scientific work is repeatable. Also, this equation is repeated as Eq. 24. To improve efficiency, please give this an equation number on first use, and avoid repetition of equations.
9. p. 18: It is stated that zero fluctuations in the along shore directions are assumed. This makes no sense - turbulence is always three dimensional, and there is no physical situation where such an assumption is justified, and Eq. 23 is not a proper estimation of k. The turbulent kinetic energy k can be approximated by one component, but this should involve a factor 1.25 (see e.g. Scott et al. 2005) rather than 0.5 in Eq. 23. Please correct this and revise accordingly.

10. Table 1: fixedValue is indicated for the velocities - which value? (Presumably this is zero, but this certainly needs to be clarified).

References

Aniel-Quiroga, I., Vidal, C., Lara, J. L., Gonzalez, M., & Sainz, A. (2018). Stability of rubble-mound breakwaters under tsunami first impact and over-flow based on laboratory experiments. Coastal Engineering, 135, 39–54. https://doi.org/10.1016/j.coastaleng.2018.01.004

Biscarini, C. (2010). Computational fluid dynamics modelling of landslide generated water waves. Landslides, 7(2), 117–124. https://doi.org/10.1007/s10346-009-0194-z

Larsen, B. E., Fuhrman, D. R., Baykal, C., & Sumer, B. M. (2017). Tsunami Induced Scour Around Monopile Foundations. Coastal Engineering, 129, 36–49. https://doi.org/10.1016/j.coastaleng.2017.08.002

Montagna, F., Bellotti, G., & Di Risio, M. (2011). 3D numerical modeling of landslide-generated tsunamis around a conical island. Natural Hazards, 58(1), 591–608. https://doi.org/10.1007/s11069-010-9689-0

Qu, K., Ren, X. Y., & Kraatz, S. (2017). Numerical investigation of tsunami-like wave hydrodynamic characteristics and its comparison with solitary wave. Applied Ocean Research, 63, 36–48. https://doi.org/10.1016/j.apor.2017.01.003

Scott, C. P., Cox, D. T., Maddux, T. B., & Long, J. W. (2005). Large-scale laboratory observations of turbulence on a fixed barred beach. Measurement Science and Technology, 16(10), 1903–1912. https://doi.org/10.1088/0957-0233/19/10/004

---

## Referee Comment (RC3) · Anonymous Referee #3 · 17 Jul 2018

General comments The main goal of the paper is to present comparisons of two hydrodynamical models. The numerical simulations are focused on tsunami wave propagation, specially at the inland inundation and impact on individual structures scales. Besides model-to-model comparisons, the performance of the models are evaluted with directional wave basin data, published elsewere. The variables analysed included the flow depth (water level), the velocity field and the momentum flux. This way two tools are now better known and can be used by planners and coastal managers. The results are interesting and useful for NHESS readers. The paper is well wrtitten overall.

[Figure]

I recommend publication with a few corrections and clarifications. Specific comments
The authors gave sound explanations concerning the NLSWE and RANS models (either directly or through the pertinent references): the mathematical basis, the options
and simplifications of the configurations adopted, the settings and the boundary and
initial conditions. The results are, basically, presented by figures. It would be important
to estimated some quantitative scores, in particular for the variables that are observed
directly (water levels and velocities). These objective measures of the performance of
each model will allow to understand their specifics. Particularly, it will be easier to separate what the authors called "near and post-impact" which, as it is right now, seems
rather arbitrary. Technical corrections Through the whole text - a comma before "and".
Pag 2 and 4 - The fourth paragraph of pag 2 (the one starting with "The scale...") is
repeated ipsis verbis in pag 4 (second paragraph). Pag 5 - Manning's coeficients are
not friction factors (nondimensional numbers). They have dimensions. The value of
the Manning coeficient should reflect the type of material of the bottom used in the
Laboratory (the simulations were at the model scale) and not justified with a reference.
Pag 16 - In the second paragraph the two steps methodology should be clarified. What
end of the domain are refering ? Pag 18 - In the sentence after equation (23) it should
be I not u' for the definition of turbulence intensity. Pag 20 - There is an inconsistency.
There is text between 4.3 and 4.3.1 (contrary to between 4.2 and 4.2.1). Pag 30 - The
first conclusion should be rephrased. The words "only" and "while" are misleading in
this context. Pag 30 - In the third conclusion what it is the meaning of "correct value" ?
It is a model-to-model comparison.

---

## Author Comment (AC1) · 19 Jul 2018

The authors would like to thank the reviewer for time and the insightful comments. We have incorporated these comments into the revised manuscript and hope that we have addressed any concerns. Specific responses to review comments are shown below.

Page 1 - 5: I would suggest if the Introduction can be tailored to a broader audience and be more concise in terms of purpose, application and scope of the paper. Most of the existing introduction can then go under a section on Previous Work. Also the authors

[Figure]

could include examples (in possibly a separate section) of existing coastal structures in tsunami inundation areas that have utilised 2D or 3D modelling studies to determine forces on structures. You might want to cite: Ingraffea, Nathan & Libby, Mark, 2015. Design of a Tsunami Vertical Evacuation Refuge Structure in Westport, Washington. Structures Congress 2015, pp.1530–1537. González, Frank, Randy LeVeque, and Loyce Adams. "Tsunami Hazard Assessment of the Ocosta School Site in Westport, WA." (2013)

The first section has been re-arranged to include two sub-sections. The scopes and goals of the paper are more explicitly introduced and summarized in section 1.1. The example above and some explicit goals of the paper are added as the last two paragraph of section 1.1.

Page 1, Line 10: The line should read, "However, it is not clear whether these equations ..."

This has been modified in the manuscript.

Page 3, Line 24: Has not the increased computing power affected both tsunami runup process and wave impact on an individual structure.

This has been addressed in the manuscript.

Page 8, Line 17: space after i,

A space has been added.

Page 9, Line 33-34: the line should read, "..., causing the measurement to oscillate dramatically."

This has been modified in the manuscript.

Page 13, Line 13: the momentum flux in equation 20 is in parenthesis so replace denominator by parenthesis

This has been modified in the manuscript.

Page 15, Line 1: delete in the experiment. The sentence already makes it clear that the sampling rate is for the experiment.

This has been modified in the manuscript.

Page 15, Line 7: Define CSZ here i.e. Cascadia Subduction Zone.

CSZ has been define earlier in the first paragraph of section 4.1.

Page 28 - 30: The conclusion may be strengthened by suggestions for the practitioner as to when might it be useful to utilise three-dimensional model studies rather than two dimensional studies in designing coastal structures within tsunami inundation areas and whether the increased computational power is really necessary or not. This point may be connected to looking back on what may have been done differently when determining forces to design for example the Tsunami Vertical Evacuation Refuge Structure in Westport.

Some suggestions have been added to the conclusion section.

Figure 3. Add legend, remove grid, add one label for time and velocity along x and y axis respectively so you can then remove Abscissa: time (s) a. Ordinates: velocity (m/s) from the caption

While we understand the intent of this comment, we feel that adding labels for time and velocity along x and y axis would create an odd aesthetic for the figure and have chosen to leave this figure as is.

Please also note the supplement to this comment:
https://www.nat-hazards-earth-syst-sci-discuss.net/nhess-2018-150/nhess-2018-150-AC1-supplement.pdf

2018-150, 2018.

**Supplement:**

[revised manuscript text omitted]

---

## Author Comment (AC2) · 19 Jul 2018

The authors would like to thank the reviewer for time and the insightful comments. We have incorporated these comments into the revised manuscript and hope that we have addressed any concerns. Specific responses to review comments are shown below.

1. As mentioned above, CFD of tsunamis are relatively rare. As this is much of the novelty of the present work, a more thorough literature review on this general topic would seem warranted, as several seemingly relevant papers are not cited. Such works

[Figure]

seemingly include: Biscarini (2010), Montagna et al. (2011), Larsen et al. (2017), as well as Aniel-Quiroga et al. (2018).

We have added those to the introduction section. Aniel-Quiroga et al. (2018) ("Stability of rubble-mound breakwaters under tsunami first impact and overflow based on laboratory experiments") is an experiment study that does not include any numerical modeling of tsunamis so we opted not to include that.

2. p. 5: B(x,y) is ambiguously defined as the topography. Does this mean the bed elevation? Please clarify.

This has been clarified in the manuscript.

3. p. 9: It is stated that z is perpendicular to the flume bottom. Would it not be simpler to state that z is vertical?

This has been updated in the manuscript.

4. p. 9: Discussing mesh resolution strictly in dimensional terms gives little physical meaning. Please also add discussion in terms of wall units, $z^+=z*U\_f/nu$, where $U\_f$ is the friction velocity and nu the kinematic fluid viscosity.

The mesh sizes in the unit of wall units have been added for both models in the manuscript in page 9.

5. p. 9: I am not convinced that simply making B(x,y) very large properly simulates a column. How exactly has this been tested? Why should the vertical column wall be modelled differently than other vertical walls?

We chose to model a column this way since we would like to make use of GeoClaw's ability of handling dry and wet cells in tsunami inundation. GeoClaw has been shown to be able to model the movement of shorelines well [1-3], during which computational cells can switch between dry and wet states. Any edge between a cell that is wet and a cell that stays dry in a time step is handled with reflecting boundary conditions and

acts as a wall. So setting the topography in the column high enough that it never gets wet is an easy way to impose wall boundary conditions around it.

The vertical walls of the flume are boundaries of the computational domain, which requires boundary conditions (in this case, they all have reflecting wall boundary conditions). In theory, the vertical column can be modeled in the same way with a modification of the code to introduce interior boundaries, but this is not necessary since the same conditions are already imposed at the interface between wet and dry cells

[1] George, D., & LeVeque, R. J. (2006). Finite Volume Methods and Adaptive Refinement for Tsunami Propagation and Inundation. PhD Thesis, 24(5), 319–328. [2]Berger, M. J., George, D. L., LeVeque, R. J., & Mandli, K. T. (2011). The GeoClaw software for depth-averaged flows with adaptive refinement. Advances in Water Resources, 34(9), 1195–1206. [3]LeVeque, R. J., George, D. L., & Berger, M. J. (2011). Tsunami modelling with adaptively refined finite volume methods. Acta Numerica, 20, 211–289.

6. Please add axes with labels to Figure 1, this will greatly improve clarity.

The x and y axes have been added.

7. Forces are estimated using a drag coefficient in Eq. 20. Why is the more general Morrison equation not used?

The Morison equation is a semi-empirical equation for the inline force on a body in oscillatory flow and is widely used in computing wave loads on offshore structures like cylindrical legs of an ocean platform. It is written as F_morison = F_inertial + F_drag = \rho * C_m * V * du/dt + 0.5 * \rho * C_d * A * u * |u|, where u = u(t) is free stream velocity and usually known from wave theory, depending on which type of wave is assumed (Linear, Stokes 3rd order, or Stokes 5th order etc.). The equation also assumes that the flow acceleration is more or less uniform at the location of the body. For instance, for a vertical cylinder in surface gravity waves this requires that the diameter of the cylinder is much smaller than the wavelength. We believe these typical

application scenarios for the Morison equation are not applicable to our problem. For example, assuming we have a building that's 1m x 1m x 1m, if we sample velocity of the fluids at a point that's 0.5 m in front of it and 0.5 m above the ground, versus at a point that's 1 m in front of the building and 0.8 m above the ground, and use time histories of these two velocities to compute the inertial force and drag force in the equation, the results can be very different for the cases we present.

Note that in the manuscript, we don't have this problem when drag force is computed, since we used a case where the building was removed and the velocity was sampled at the center of the building. The history of this velocity cannot be used to compute the inertial force in Morison equation since there is no building resisting the flow thus du/dt > 0 during most of the time (until the peak has passed the location of interest). Again, we believe although this semi-empirical equation is used conventionally in ocean and petroleum engineering, it is not suitable in current scenario.

In addition to the discussion above, using drag coefficient to estimate the hydrodynamic forces that contributes to tsunami loads on structures is what's recommended by FEMA P-646 [1] as well as by the chapter for tsunami loads and effect for coastal structures in the latest ASCE 7-16 [2]. These practices have been used for the tsunami hazard study community and recently, applied to the design of the first tsunami vertical evacuation structure in the United States [3].

[1] Applied Technology Council: Guidelines for Design of Structures for Vertical Evacuation from Tsunamis. Second Edition (FEMA P-646), FEMA P-646 Publication, https://doi.org/10.1061/40978(313)7, 2012 [2] American Society of Civil Engineers (ASCE): Minimum Design Loads for Buildings and Other Structures, Standard ASCE/SEI 7-16, 2016 [3] Ash, C.: Design of a Tsunami Vertical Evacuation Refuge Structure in Westport, Washington, in: Structures Congress 2015, pp. 1530–1537, 2015

8. p. 15: The function s(t) is given, but without specifying parameters A and beta,

hence the reader is given no information regarding the duration. Please clearly define these parameters. Sufficient information must always be given such that scientific work is repeatable. Also, this equation is repeated as Eq. 24. To improve efficiency, please give this an equation number on first use, and avoid repetition of equations.

This has been addressed accordingly in the manuscript.

9. p. 18: It is stated that zero fluctuations in the along shore directions are assumed. This makes no sense - turbulence is always three dimensional, and there is no physical situation where such an assumption is justified, and Eq. 23 is not a proper estimation of k. The turbulent kinetic energy k can be approximated by one component, but this should involve a factor 1.25 (see e.g. Scott et al. 2005) rather than 0.5 in Eq. 23. Please correct this and revise accordingly.

We have corrected this in the manuscript. We did not apply the factor 1.25 before since the same approach is used in [1].

[1] Lin, P. and Liu, P. L. F.: A numerical study of breaking waves in the surf zone, Journal of Fluid Mechanics, 359, 239–264, https://doi.org/10.1017/S002211209700846X, 1998.

10. Table 1: fixedValue is indicated for the velocities - which value? (Presumably this is zero, but this certainly needs to be clarified).

Yes, for the fixedValue boundary condition, a constant value of 0 is used for the model in this study. This has been added to the manuscript.

Please also note the supplement to this comment:
https://www.nat-hazards-earth-syst-sci-discuss.net/nhess-2018-150/nhess-2018-150-AC2-supplement.pdf

**Supplement:**

[revised manuscript text omitted]

---

## Author Comment (AC3) · 19 Jul 2018

The authors would like to thank the reviewer for the time and the insightful comments that were provided. We have incorporated these comments into the revised manuscript and hope that we have addressed any concerns. Specific responses to review comments are shown below.

Specific comments

The authors gave sound explanations concerning the NLSWE and RANS models (ei-

ther directly or through the pertinent references): the mathematical basis, the options and simplifications of the configurations adopted, the settings and the boundary and initial conditions.

The results are, basically, presented by figures. It would be important to estimate some quantitative scores, in particular for the variables that are observed directly (water levels and velocities). These objective measures of the performance of each model will allow to understand their specifics. Particularly, it will be easier to separate what the authors called "near and post-impact" which, as it is right now, seems rather arbitrary.

This is a good point and we considered a variety of ways to compare the two approaches. In several iterations we presented quantitative representations of the results, but ultimately decided to remove them since many of the results (e.g discrepancies between numerical prediction and experimental measurements) lack quantitative consistency over time and between gauges. We feel that showing and discussing the results qualitatively with figures is more appropriate than giving quantitative conclusions because (1) the qualitative conclusions are quite consistent between gauges, and (2) we would expect quantitative differences to be site specific, while the qualitative results would be expected to be consistent from site to site.

Technical corrections

Through the whole text - a comma before "and".

We have re-checked the manuscript and added comma before "and" where it was grammatically appropriate.

Pag 2 and 4 - The fourth paragraph of page 2 (the one starting with "The scale...") is repeated ipsis verbis in page 4 (second paragraph).

This has been addressed in the manuscript.

Pag 5 - Manning's coefficients are not friction factors (nondimensional numbers). They have dimensions. The value of the Manning coeficient should reflect the type of material of the bottom used in the Laboratory (the simulations were at the model scale) and not justified with a reference.

The unit for Manning's coefficient has been added. 0.025 is a very typical value used in tsunami simulation to represent earth ground. In our study, 0.025 is shown to give good agreement in arrival time of tsunami waves.

Pag 16 - In the second paragraph the two steps methodology should be clarified. What end of the domain are refering ?

This has been clarified in the manuscript.

Pag 18 - In the sentence after equation (23) it should be I not u' for the definition of turbulence intensity.

We believe we are referring to I for the definition of turbulence intensity there.

Pag 20 - There is an inconsistency. There is text between 4.3 and 4.3.1 (contrary to between 4.2 and 4.2.1).

The text (discussion) between 4.3 and 4.3.1 are motivated by the necessity of introducing flow parameters and gauges in the experiment. There is no text between 4.2 and 4.2.1 because we feel that there is not much common between the setup of two models and we separate the discussion of the setup for two model immediately at the beginning of section 4.2. We feel that it is acceptable to either have or have no text between a section title and the subsections within that section.

Pag 30 – The first conclusion should be rephrased. The words "only" and "while" are misleading in this context.

This has been modified in the manuscript.

Pag 30 - In the third conclusion what it is the meaning of "correct value" ? It is a model-to-model comparison.

[Figure]

The word "correct value" has been replaced with a more appropriate phrase.

Please also note the supplement to this comment:
https://www.nat-hazards-earth-syst-sci-discuss.net/nhess-2018-150/nhess-2018-150-AC3-supplement.pdf

**Supplement:**

[revised manuscript text omitted]

---

## Author Response (AR1)

Dear editor:

We have attached a modified version of the manuscript and a marked-up version that highlights the changes we have made.

We have included comments of the referees that we think are appropriate. Details can be found in our replies during the interactive discussion.

Best,
Xinsheng